# Automatic Modulation Classification Based on Deep Feature Fusion for High Noise Level and Large Dynamic Input

**DOI:** 10.3390/s21062117

**Published:** 2021-03-17

**Authors:** Hui Han, Zhiyuan Ren, Lin Li, Zhigang Zhu

**Affiliations:** 1State Key Laboratory of Complex Electromagnetic Environment Effects on Electronics and Information System (CEMEE), Luoyang 471003, China; cemee_hanhui@163.com; 2School of Electronic Engineering, Xidian University, Xi’an 710071, China; zhiyuan_xd@163.com (Z.R.); zgzhu@xidian.edu.cn (Z.Z.)

**Keywords:** automatic modulation classification, stacked auto-encoder, convolutional neural network, probabilistic neural network

## Abstract

Automatic modulation classification (AMC) is playing an increasingly important role in spectrum monitoring and cognitive radio. As communication and electronic technologies develop, the electromagnetic environment becomes increasingly complex. The high background noise level and large dynamic input have become the key problems for AMC. This paper proposes a feature fusion scheme based on deep learning, which attempts to fuse features from different domains of the input signal to obtain a more stable and efficient representation of the signal modulation types. We consider the complementarity among features that can be used to suppress the influence of the background noise interference and large dynamic range of the received (intercepted) signals. Specifically, the time-series signals are transformed into the frequency domain by Fast Fourier transform (FFT) and Welch power spectrum analysis, followed by the convolutional neural network (CNN) and stacked auto-encoder (SAE), respectively, for detailed and stable frequency-domain feature representations. Considering the complementary information in the time domain, the instantaneous amplitude (phase) statistics and higher-order cumulants (HOC) are extracted as the statistical features for fusion. Based on the fused features, a probabilistic neural network (PNN) is designed for automatic modulation classification. The simulation results demonstrate the superior performance of the proposed method. It is worth noting that the classification accuracy can reach 99.8% in the case when signal-to-noise ratio (SNR) is 0 dB.

## 1. Introduction

Automatic modulation classification is a technology for the automatic classification of signal modulation types [1], which is widely used in interference identification, spectrum sensing, electronic countermeasures and other fields. With the significant development of radio communication, the signal modulation and electromagnetic environment are becoming increasingly complex and diverse, and result in a high noise level, usually greater than −100 dBHz, viz. SNR is extremely low. Meanwhile, due to the low probability of the intercept (LPI), the SNR of the intercepted signal is constantly changing in the dynamic range, and therefore the range of SNR becomes much larger. These new circumstances make AMC tasks more difficult. Hence, it is crucial to explore more effective techniques for AMC.

Generally, AMC approaches can be classified into two categories: likelihood-based (LB) methods and feature-based (FB) methods. The likelihood-based method, based on likelihood function and combined with a hypothesis test, can obtain the highest classification performance in theory. The likelihood function can be divided into three groups: hybrid likelihood ratio test [2], average likelihood ratio test [3], and generalized likelihood ratio test [4]. However, such methods suffer from high computational complexity due to too many unknown parameters. On the contrary, feature-based methods consisting of feature extraction and classifier design, have been widely applied in the AMC field to recognize more modulation types with lower algorithm complexity (see [5]). Consequently, one or more stable and distinguishing features can improve the classification accuracy significantly. In recent years, researchers have studied various features such as high-order cumulants [6,7,8], instantaneous features [9], cyclostationary features [10], etc., and a variety of methods for classification. Besides the decision tree model, the existing classification methods are mainly based on machine learning algorithms, including the artificial neural network (ANN) [11], support vector machine (SVM) [12], and SAE [13].

With the rapid development of artificial intelligence (AI) technology, deep learning (DL), with its excellent data-processing ability, has attracted widespread attention and been used in various fields. In recent years, researchers have applied related algorithms in AMC, and achieved expected results. (see [14,15,16]). Experiments in [14] show that the classification accuracy of a convolutional neural network trained by in-phase/quadrature (I/Q) data is superior to that trained by cyclic-moment features. Reference [15] proposed a two-step training method based on transfer learning. The results show that the training speed of this method is from 40 to 1700 times faster than that of single-step training. In [16], an AMC algorithm using CNN to extract constellation features of digital communication signals is studied, which combines image classification and deep learning.

In this paper, an AMC scheme based on deep learning and feature fusion is proposed, which can achieve high classification accuracy in a low SNR environment, and within a large range of SNR. In [17], a CNN-based fusion scheme is proposed to deal with the problem of signal duration burst. In [18], an image fusion algorithm is proposed for modulation classification, which considers the images from different time-frequency methods. Recently, the multimodal deep learning approaches that proposed image and speech recognition have also been considered for radio signals. Differing from the existing feature fusion approaches, we utilized the complementarity among features in different domains, and especially, for high noise level and large dynamic input. Specifically, the contributions of the paper are as follows:

(i) For frequency features, we consider the differences between spectrum obtained by FFT and power spectrum by Welch approach (windowed smoothing). The spectrum with high-frequency resolution preserves the detailed information of the modulation characteristics. On the other hand, the windowed smoothing power spectrum with low-frequency resolution presents the stable feature, which is suitable for low SNR;

(ii) CNN and SAE are proposed for feature extraction. A new CNN scheme is constructed to extract sparse features from the spectrum. Meanwhile, the unsupervised SAE is proposed for feature extraction from the Welch power spectrum;

(iii) The statistical feature is proposed for complementarity, which stands for the information at the time domain. Specifically, we extract the statistical features from instantaneous amplitude and instantaneous phase, and combine the higher-order cumulants;

(iv) A PNN scheme is proposed for modulation classification, which provides a much higher classification accuracy than the existing methods.

The rest of this paper is arranged as follows. Section 2 introduces the signal model and dataset used in this paper. Section 3 describes the new framework of the proposed method, including feature fusion and classification model. Section 4 shows simulation results and analysis. A brief conclusion is given in Section 5.

## 2. Signal Model

In this paper, we consider the additive white Gaussian noise (AWGN) channel for modulation signals. Formally, an intercepted (received) radio signal r(t) can be expressed as
(1)$$r\left(t\right)=s\left(t\right)+n\left(t\right),$$
where s(t) and n(t) denote the transmitted signal and the Gaussian noise, respectively; see details in [12].

When transmitted signal is M-ASK, M-FSK or M-PSK (see Table 1 for the modulation types), *s*(*t*) can be represented as
(2)$$s\left(t\right)=\left[A_{m}\sum_{n}a_{n}g\left(t-nT_{s}\right)\right]\cos\left(2\pi \left(f_{c}+f_{m}\right)t+\phi_{0}+\phi_{m}\right),$$
where $A_{m}$ and $a_{n}$ represent the modulation amplitude and symbol sequence, respectively. Note that $a_{n}$ is modulated by the window function $g\left(t\right)$, defined as
(3)$$g\left(t\right)=\left\{\begin{array}{c} 1,\;\;\;\;1\leq t\leq T_{s}\\ 0.\;\;\;\;others \end{array}\right.$$
$T_{s}$ represents the symbol period, $f_{c}$ and $f_{m}$ denote carrier frequency and modulation frequency.

For signal with M-QAM, there are two orthogonal carriers, modulated by the amplitudes of $a_{n}$ and $b_{n}$, written as
(4)$$s\left(t\right)=\left[A_{m}\sum_{n}a_{n}g\left(t-nT_{s}\right)\right]\cos\left(2\pi f_{c}t+\phi_{0}\right)+\left[A_{m}\sum_{n}b_{n}g\left(t-nT_{s}\right)\right]\sin\left(2\pi f_{c}t+\phi_{0}\right).$$

In this paper, we mainly consider common digital modulations, such as those described above. We define the modulation dataset *S*, used to test our method, where $S=\{S^{\left(c\right)},c=1,2,\ldots ,C\}$, *C* is the number of the modulation types. Table 1 shows the detailed dataset *S*, used in our paper, where S= {2PSK, 4PSK, 2ASK, 4ASK, 8ASK, 2FSK, 4FSK, 8FSK, 32QAM, 64QAM}. Note that we consider 10 modulation types.

## 3. The Proposed Framework

### 3.1. Overview of the Proposed Framework

It can be observed from existing works on AMC, that features in different domains, such as time, frequency, cyclostationary, high-order cumulants, and time-frequency representation, have been studied, analyzed and tested broadly. However, few researchers have attempted to explore the differences among features from different domains, and integrate different features with complementary advantages. In this section, first we present the proposed feature-fusion framework for modulation classification; see Figure 1. The framework contains three steps: signal processing, feature extraction and classification.

The FFT and Welch power spectrum [19] are used to convert all signals into the frequency-domain, but preserving different information. Then, we introduce a self-designed CNN model to learn the features after FFT. SAE is used to extract unsupervised features from Welch spectrum data. Meanwhile, features based on instantaneous characteristics and higher-order cumulants are used as statistical features. After that, in order to obtain better feature representation, all the extracted features are combined to form a joint feature vector. Finally, to further improve the classification performance, we utilize a PNN for final classification, which has a higher classification accuracy and better fitting effect than BP neural network [20]. Figure 2 shows the sketch map of the proposed method.

### 3.2. Features Extraction

#### 3.2.1. CNN-Based Feature Extraction

Deep learning, as a novel development method, can extract more meaningful features through its hierarchical learning process. For modulation classification, DL-based methods can automatically learn distinctive representations of high-dimensional data, such as the received radio signals [21].

CNN is composed of a convolution layer and pooling layer, so as to extract effective features. The basic operation of the convolution layer and pooling layer is shown in Figure 3. There are usually two kinds of pooling operations: one is max-pooling, where the maximum value is selected as the output value in the size coverage area of the pooling filter. The other is mean-pooling, which pools the average value of the coverage area as the output value. Mean-pooling and max-pooling correspond to mean filtering and maximum filtering in image processing, and can simultaneously reduce the number of features.

The activation function in the convolution layer is the rectified linear unit (ReLU), which can be expressed as
(5)$$ReLU\left(x\right)=\max\left\{0,x\right\}$$

The neuron will enter a “dead” state and is insensitive to other gradients when a large gradient flows through a neuron, which is called the “dying ReLU” [22]. To avoid the “dying ReLU”, Swish function [23] and SeLU function [24] are selected as the activation function in the convolution layer and the fully connected layer, which are expressed as (Equation 6) and (Equation 7), respectively.
(6)$$Swish\left(x\right)=x\cdot \mathrm{sigmoid}\left(x\right)$$
(7)$$SeLU\left(x\right)=\lambda \left\{\begin{array}{c} x,\;\;\;\;\;\;\;\;x>0\\ \alpha e^{x}-\alpha ,x\leq 0 \end{array}\right.$$

Due to the dimension of the input data being different from that of the traditional images or videos, we set up a new CNN structure. This structure consists of 16 layers, including a convolution layer, pool layer, full connection layer and Softmax layer. Table 2 shows the detailed model configuration.

In this paper, we use a 1 × 3 convolution kernel for feature extraction. We also introduce a 1 × 2 pool core to reduce the parameters. In order to avoid over-fitting of the network, a dropout operator with ratio of 0.5 is introduced into the convolution layer. Moving average and L2 regularization are introduced into the model. Finally, the last layer of the network is connected to the Softmax classifier to output the prediction score of the modulation signal types.

#### 3.2.2. SAE-Based Feature Extraction

The Welch power spectra of M-PSK and M-QAM signals are relatively close to each other [19], which means that supervised learning may cause feature confusion. Hence, we introduce the SAE method, a simple neural network for feature extraction, which is an unsupervised learning approach.

SAE first takes a non-linear mapping of the input $x\in R^{d_{x}}$ into a hidden representation $y\in R^{d_{y}}$, where $d_{x}$ and $d_{y}$ are the dimensions of *x* and *y*, respectively. Then, map *y* back to the input space as $x^{\prime }$. These two steps are called the encoding and decoding, respectively [25],
(8)$$y=h\left(Wx+b\right),$$
(9)$$x^{\prime }=h^{\prime }\left(W^{\prime }y+b^{\prime }\right).$$

Here, *h* stands for the encoder transfer function, where the commonly used function is Log-sigmoid function (Logsig), written as
(10)$$\mathrm{Log}\mathrm{sig}\left(x\right)=\frac{1}{1+e^{-x}}.$$
$h^{\prime }$ is the decoder transfer function, which can be linear (Purelin). *W* and $W^{\prime }$ are the weight matrices of the encoder and decoder, respectively. *b* and $b^{\prime }$ are the bias units.

The pre-training of each auto-encoder aims to adjust the weight matrices by minimizing the cost function in (Equation 11) with respect to the weight matrices *W* and $W^{\prime }$ and biases *b* and $b^{\prime }$.
(11)$$Cost=∥x-x^{{}^{\prime }}{∥}_{2}^{2}+\frac{\lambda }{2}\sum_{l=1}^{L}\sum_{j=1}^{n}\sum_{i=1}^{k}{\left({\left(W\right)}_{ij}^{l}\right)}^{2}+\beta \psi_{s},$$
where *L* is the number of the hidden layers, *n* is the number of data samples, *k* is the number of input features, λ and β are coefficients. $\psi_{s}$ is the sparsity regularizing term given by the Kullback–Leibler divergence.

The Cost function includes three terms, where the Cross Entropy function (the first term) fits the input data, the weight-regularizing term (the second term) is controlled by the coefficient λ and the sparsity-regularizing term (the third term) is controlled by the coefficient β.

In order to reveal the unsupervised learning of SAE, the Welch power spectrum, extracted features and decoding outputs of four digital modulation signals, 2PSK, 32QAM, 4FSK and 8FSK are shown in Figure 4. The top row of Figure 4 shows the input of the SAE, the Welch spectrum of the signal in (Equation 1) with SNR = 0 dB. The middle row shows the SAE feature vectors used for classification. The bottom row shows the decoding of unsupervised learning. Observe that the output of SAE is very close to the input. Actually, the root–mean–square error (RMSE) between the input and the output is negligible, which indicates that the loss of feature extracted by the self-encoder can be ignored when SNR = 0 dB.

#### 3.2.3. Statistical Feature Extraction

Considering the complementary information of time-domain, the instantaneous amplitude (phase) statistics and higher-order cumulants are extracted as the statistical features for fusion, which can reveal statistical time-varying information [26]. Meanwhile, the higher-order cumulants are insensitive to Gaussian noise and robust to phase rotation, see [27] and [28].

Suppose the received signal x(n) (after the quadrature demodulation and normalization, with complex value and zero-mean) is written as
(12)$$x\left(n\right)=x_{I}\left(n\right)+jx_{Q}\left(n\right)=a\left(n\right)e^{j\phi (n)},n=1,2,\ldots ,N_{s},$$
where $x_{I}\left(n\right),x_{Q}\left(n\right)\in \left[-1,1\right]$ are in-phase and quadrature components, and a(n)∈[0,1],ϕ(n)∈[0,2π] are called the instantaneous amplitude and phase, respectively. $N_{s}$ is the number of samples. Note that the means of $x_{I}\left(n\right)$ and $x_{Q}\left(n\right)$ are equal to 0, due to the fact that x(n) is zero-mean.

Let $\tilde{a}\left(n\right)$ and $\tilde{\phi }\left(n\right)$ be the centered instantaneous amplitude and phase, respectively, defined as
(13)$$\tilde{a}\left(n\right)=a\left(n\right)-\bar{a(n)},$$
(14)$$\tilde{\phi }\left(n\right)=\phi \left(n\right)-\bar{\phi (n)},$$
where $\bar{x}$ denotes the mean of x. Then, we define the following statistics.

(a) The periodic characteristic of $\tilde{a}\left(n\right)$
(15)$$\gamma_{max}=\frac{\max{\left|F[\tilde{a}\left(n\right)]\right|}^{2}}{N_{s}},$$
where F stands for the $N_{s}$-point discrete Fourier transform of $\tilde{a}\left(n\right)$.

(b) The standard deviation of $\tilde{a}\left(n\right)$
(16)$$\sigma_{a}={∥\tilde{a}\left(n\right)∥}_{2}.$$

(c) The standard deviations of ϕ(n) and $\tilde{\phi }\left(n\right)$
(17)$$\sigma_{\phi }={∥\phi \left(n\right)∥}_{2},\;\;\sigma_{\tilde{\phi }}={∥\tilde{\phi }\left(n\right)∥}_{2}.$$

(d) The standard deviation of the derivative of $\phi_{u}\left(n\right)$
(18)$$\sigma_{f}={∥\phi_{d}\left(n\right)∥}_{2},$$
where $\phi_{d}\left(n\right)=\phi_{u}^{\prime }\left(n\right)$, $\phi_{u}\left(n\right)$ is the unwrapped ϕ(n).

Next, we introduce the 4th-, 6th-, and 8th-order cumulants of the received signal x(n) in (Equation 12). First, define the *p*th-order moment as
(19)$$M_{pq}=E\left[x{\left(n\right)}^{p-q}x^{*}{\left(n\right)}^{q}\right].$$

Then we have the following cumulants.
(20)$$C_{40}=M_{40}-3M_{20}^{2},$$
(21)$$C_{41}=M_{41}-3M_{20}M_{21},$$
(22)$$C_{42}=M_{42}-{\left|M_{20}\right|}^{2}-2M_{21}^{2},$$
(23)$$C_{60}=M_{60}-15M_{40}M_{20}+30M_{20}^{3},$$
(24)$$C_{63}=M_{63}-6M_{20}M_{41}-9M_{21}M_{42}+18M_{20}^{2}M_{21}+12M_{21}^{3},$$
(25)$$C_{80}=M_{80}-35M_{80}^{2}-28M_{60}M_{20}+420M_{20}^{2}M_{40}-630M_{20}^{4}.$$

Finally, we construct the features mentioned above to a vector, as follows
(26)$$\zeta =\left[\gamma_{max},\sigma_{a},\sigma_{\phi },\sigma_{\tilde{\phi }},\sigma_{f},C_{40},C_{41},C_{42},C_{60},C_{63},C_{80}\right],$$
where ζ denotes the statistical feature space.

### 3.3. Classification with PNN

In order to obtain a better signal representation, all the extracted features in Section 3.2 are combined to form a joint feature vector. To further improve the classification performance of AMC, we introduce a PNN network for final classification, see (Figure 5). The PNN network in Figure 5 is a feed-forward network [29], which realizes the prediction classification of models through Bayesian decision theory. In our proposed framework, joint features are first input into the input layer. Then, the samples are trained by the RBF neurons in the pattern layer. The similar output values are given by the summation layer, and finally, the competing neurons of output layer are judged to obtain the types of digital modulation.

## 4. Experiment Results and Analysis

In this section, we will use a series of simulations to evaluate the performance of our proposed framework. Recalling the dataset *S* (modulation types) given by Table 1, here, we set up the signal parameters as follows; see Table 3. The bandwidths of all the signals vary from 4 to 6 MHz randomly. During training, we generated 84,000 signals for all the modulation types, with SNR continuously randomly chosen from −20 to 20 dB. In addition, 21,000 examples are randomly generated with SNR ranging from −20 to 20 dB at an interval of 2 dB (with random selection of modulation type and bandwidth) to test the performance of our method. Considering the background noise −100 dBHz (dBHz is the unit of power spectral density), bandwidth 5 MHz and the sampling rate 50 MHz, the input SNR from −20 to 20 dB is equivalent to the input signal power of from −70 to −30 dBm (here, dBm is the unit of power) approximately. Hence, we consider the input dynamic range to be about 40 dB. Such a large dynamic input for training is scarcely considered in the existing works about AMC.

Firstly, in order to verify the effectiveness of feature fusion, we compare the performances of different methods; see the results in Figure 6. Note that “F”, “W” and “S” are the initials of the three features, which are CNN-based features (frequency features), SAE-based features (Welch power spectrum features) and statistical features, respectively. FW, FS and WS stand for the fusions of two types of features, chosen from “F”, “W” and “S”, with different combinations. Note that FWS stands for the fusion of all the three types of feature, which is the method proposed in this paper. We use PNN for classification, but with different schemes to fit the different sizes of input feature.

For single-type features, the classification performance of CNN-based features is much better than other two types under high SNR (when SNR ≥−2 dB), due to the global optimization of model training. However, the SAE-based features are better for low SNR (when SNR ≤−4 dB). By feature fusing, the methods FW, FS and FWS can improve the overall classification accuracy. However, the WS method is best for the case of extremely low SNR (when SNR≤−14 dB). The results show there is a significant complementarity among features, namely, the complementarities of different domains and different SNRs (low SNR and high SNR, since we consider the large dynamic range input). It is worth noting that the total classification accuracy of FWS is the highest, which proves the validity of our feature fusion frame.

In addition, Figure 7 shows the confusion matrices of the feature fusion methods above, under the condition of low SNR, namely SNR =− 6 dB. Obviously, the two M-PSK signals and the three M-ASK signals can not be distinguished by the WS, respectively. This is due to the fact that the Welch spectra of these signals are close to each other. However, our proposed method, the FWS, achieves the highest classification accuracy. This is because the combination of features in different domains achieves complementary advantages.

To demonstrate the superiority of our proposed algorithm, Figure 8 shows the classification accuracy of different methods, including the feature fusion method proposed in this paper (FWS), CNN-LSTM with the signal waveform in [14] and the three convolutional neural networks for feature and decision fusion (CNN3) in [17]. Obviously, the classification performance of FWS is superior to the other two methods at low SNRs, which proves the superiority of the proposed method for large dynamic input.

We also provide the classification accuracy of each signals by the proposed FWS method, see Figure 9. The classification accuracy of most signals is above 96% when SNR = 0 dB, and some of them even reach 100%. Although the noise has a direct impact on the signal, FSK signals also show excellent performance at low SNR (SNR =−16 dB). The average classification accuracy of M-ASK signals is comparatively lower than the other signals, but it is still greater than 70% when SNR =−8 dB.

Finally, in order to demonstrate the superiority of our classification scheme, we compare PNN with SAE [13], SVM [12] and ANN [11]. As expected, we can see from Figure 10 that the classification accuracy based on PNN is higher than the other three classifiers, especially at low SNRs, which is mainly due to the strong nonlinear approximation ability of PNN. Note that SVM is close to SAE, and the ANN model has the lowest classification rate. The results show that the proposed framework with PNN has high robustness in a complex noise environment.

Moreover, we evaluate the computational complexity of the four classifiers above by the average training time. Note that all methods are performed on the same hardware system. As shown in Table 4, the training time of SVM is much longer than other methods, which shows that SVM is not suitable for training large-scale datasets. Meanwhile, the training time of PNN is less than SAE, ANN and SAE, which is due to that PNN is easier to converge than other classifiers.

## 5. Conclusions

This paper deals with the problem of feature fusion for the complex electromagnetic environment, especially for a high noise level and large dynamic input. An AMC framework based on deep learning is proposed, which aims to fuse different features of the input signals to obtain more differentiating features. For the CNN-based and SAE-based features, the statistical features are fused and a PNN is designed for automatic modulation classification. The simulation results demonstrate the superiority of the proposed method, especially in a complex noise environment, with SNR ranges from −20 to 20 dB. In future works, more modulation types should be considered to further improve the generalization and robustness of the proposed method.

## Figures and Tables

**Figure 1 sensors-21-02117-f001:**
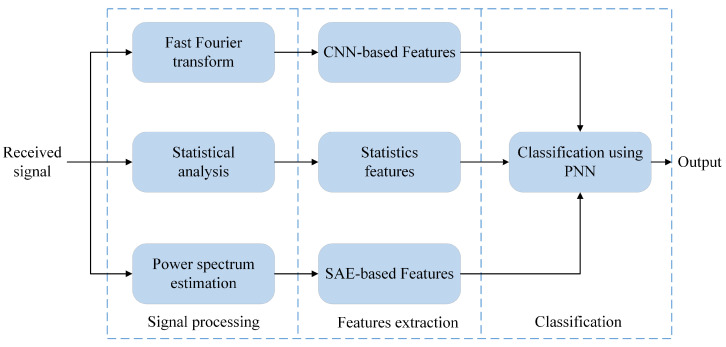
Block diagram of proposed scheme.

**Figure 2 sensors-21-02117-f002:**
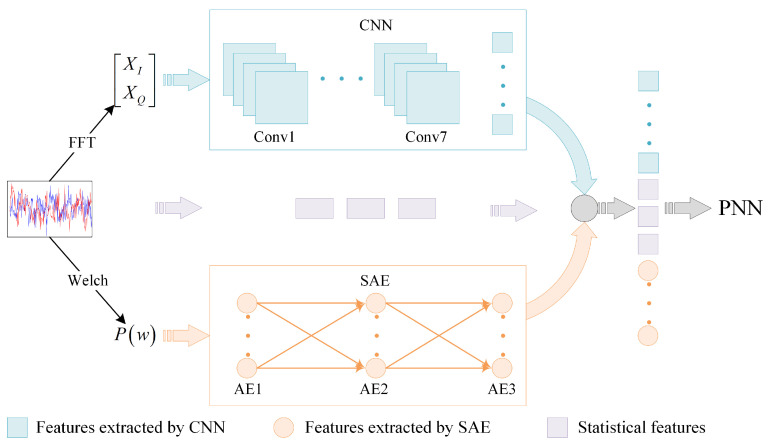
Sketch map of the proposed method.

**Figure 3 sensors-21-02117-f003:**
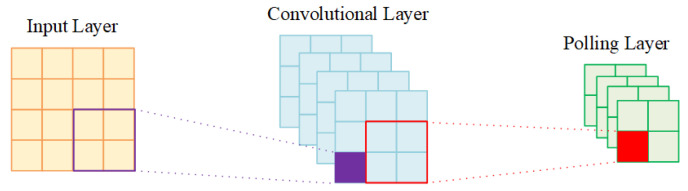
A complete convolution operation.

**Figure 4 sensors-21-02117-f004:**
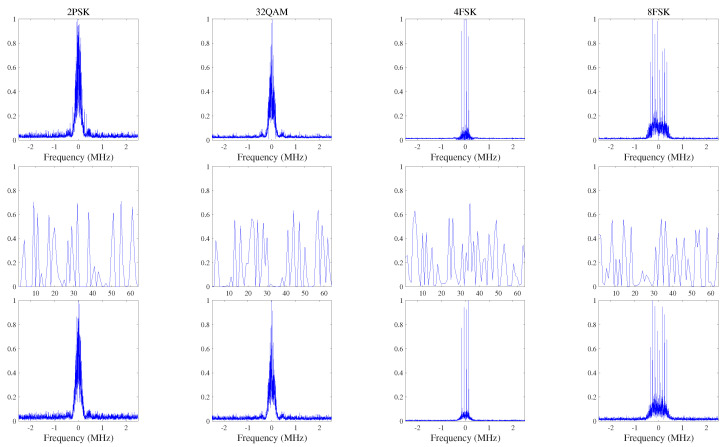
Welch spectrums, SAE features and decoding results under white Gaussian noise (SNR = 0 dB).

**Figure 5 sensors-21-02117-f005:**
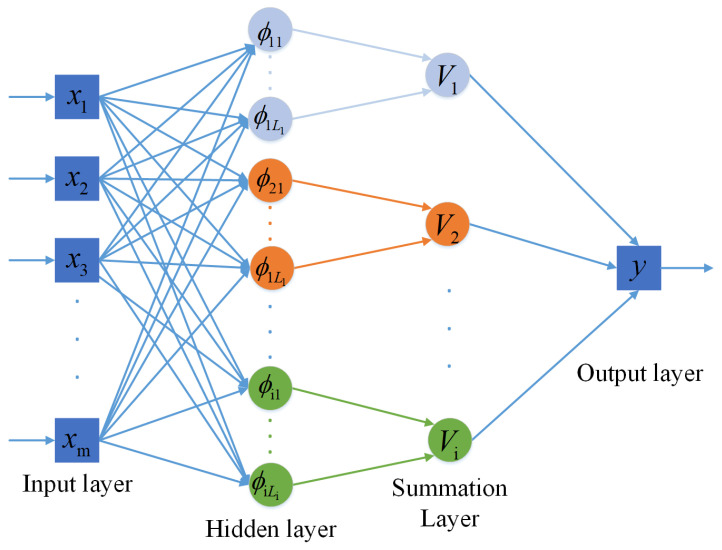
Structure of the PNN network.

**Figure 6 sensors-21-02117-f006:**
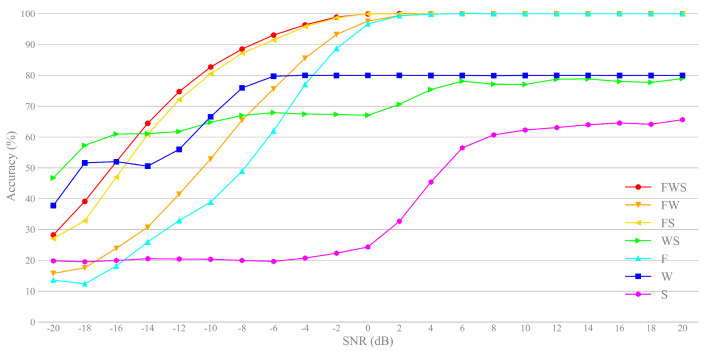
Classification accuracy vs. SNR for different features.

**Figure 7 sensors-21-02117-f007:**
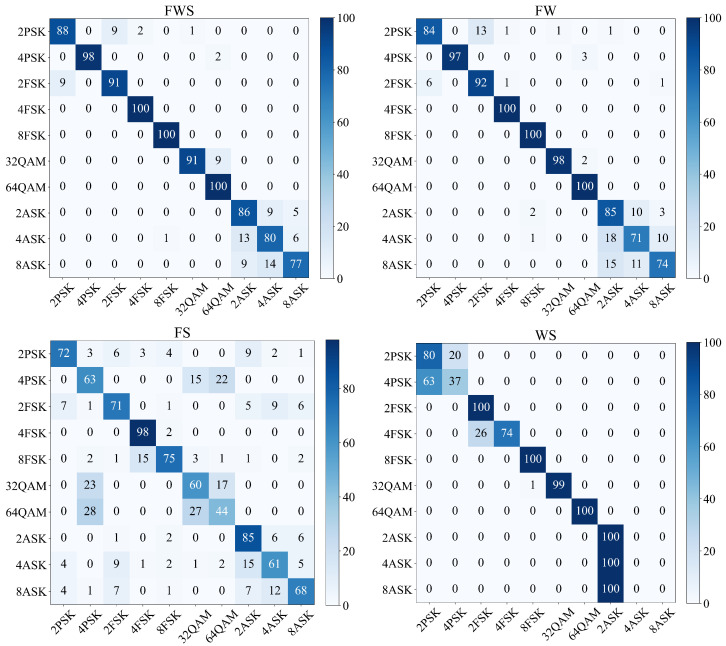
Confusion matrix of different methods when SNR =−6 dB.

**Figure 8 sensors-21-02117-f008:**
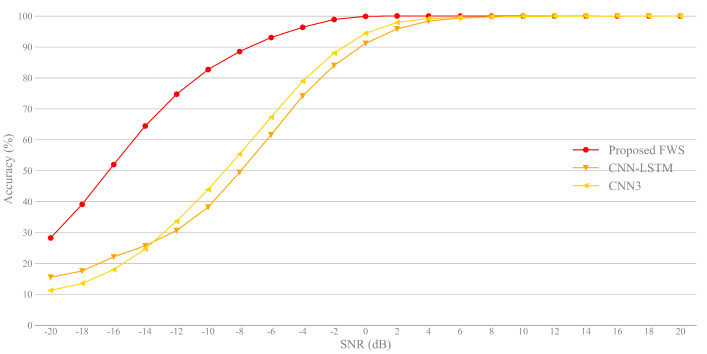
Classification accuracy vs. SNR for different methods.

**Figure 9 sensors-21-02117-f009:**
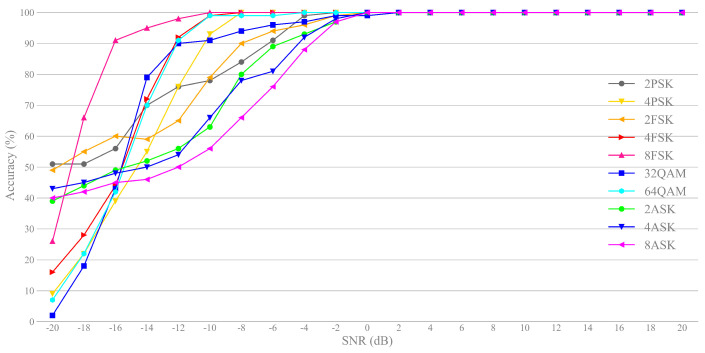
Classification accuracy vs. SNR for each signals using FWS approach.

**Figure 10 sensors-21-02117-f010:**
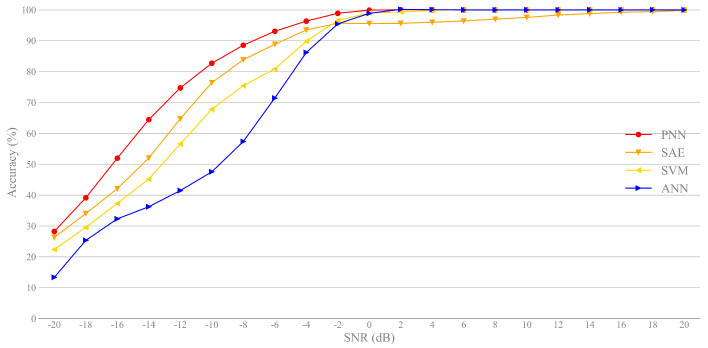
Comparison of different classifiers using FWS approach.

**Table 1 sensors-21-02117-t001:** Modulation types in dateset *S*.

Class	Name	Modulation
$S^{1}$	2PSK	2-ary Phase Shift Keying
$S^{2}$	4PSK	4-ary Phase Shift Keying
$S^{3}$	2ASK	2-ary Amplitude Shift Keying
$S^{4}$	4ASK	4-ary Amplitude Shift Keying
$S^{5}$	8ASK	8-ary Amplitude Shift Keying
$S^{6}$	2FSK	2-ary Frequency Shift Keying Keying
$S^{7}$	4FSK	4-ary Frequency Shift Keying Keying
$S^{8}$	8FSK	8-ary Frequency Shift Keying Keying
$S^{9}$	32QAM	32-ary Quadrature Amplitude Modulation
$S^{10}$	64QAM	64-ary Quadrature Amplitude Modulation

**Table 2 sensors-21-02117-t002:** CNN structure and the simulation parameters.

Layer (Type)	Output Dimensions	Parameters Number
Input	2 × 16,384	0
Conv2- Pool2/Swish	1 × 8192 × 1	48
Conv2- Pool2/Swish	1 × 4096 × 4	48
Conv2- Pool2/Swish	1 × 2048 × 4	48
Conv2- Pool2/Swish	1 × 1024 × 4	48
Conv2- Pool2/Swish	1 × 512 × 4	48
Conv2- Pool2/Swish	1 × 256 × 4	48
Conv2- Pool2/Swish	1 × 128 × 4	48
Full Connected/SeLU	64	32,768
Softmax	10	640

**Table 3 sensors-21-02117-t003:** Signal parameters.

Parameter	Value/Range	Description
Sampling frequency	50 MHz	The sampling rate
Signal length	16,384	The number of sampling points
Bandwidth	[4,6] MHz	Randomly selection
SNR for training	[−20,20] dB	The dynamic range
SNR for test	{−20,−18,…,20} dB	For evaluation
Training samples	84,000	The total samples for training
Test samples	21,000	The total samples for test
Test samples for each SNR	100	For each signal type

**Table 4 sensors-21-02117-t004:** Training times of different classifiers.

Models	PNN	ANN	SAE	SVM
Training Time (s)	13.711	18.233	34.249	231.178

## Data Availability

All modulation data and code will be made available on request to the correspondent author’s email with appropriate justification.

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
