# Peer review of "Automatic Modulation Classification Based on Deep Feature Fusion for High Noise Level and Large Dynamic Input"

_sensors, 2021, doi:10.3390/s21062117_

Round 1
Reviewer 1 Report
- Figure 4 label, instead of 4FSK it is written 44SK. The description written on line 163 to 165 should also be written on the figure 4 description.
- You compared different classifiers in terms of accuracy, but you didn’t compare them in terms of power consumption. Your proposed method performs better in terms of accuracy, but it should be clear at what cost.
Reviewer 2 Report
Review on “Automatic Modulation Classification Based on Deep Feature Fusion for High Noise Level and Large Dynamic Input”
This paper presents an AMC framework based on deep learning and highlights the possibility to fuse different features extracted from the input signals in order to achieve superior performances over the methods that are already in used nowadays. The proposed method is validated by simulation results. The paper is well written and may be published, but not before some issues are addressed.
Major concerns:
- Page 1: Try to rephrase lines 27-29:” Meanwhile, …, more than 60dB”. Some small misunderstandings may occur.
- Page 4, line 119: you mention BP neural network. Please provide a short description and explain why the actual PNN is superior.
- Page 5, line 141: “In order to avoid the death of ReLU” , what does it mean the death of ReLU? Please provide the mathematical description of ReLU function
- Page 5, line 142: Please provide the mathematical description of Swish function
- Page 5, line 143: please describe the mathematical representation of the softmax classifier
- Page 6, line 151: Please provide the mathematical description of Log-sigmoid function (Logsig) (here or in some annex)
- Page 6, figure 4: what are the axis representations on Oy axis?How did you compute the SNR, please provide the mathematical description of your parameters
- Page 7, Equation (14): why you have not used the notation instead of . The same for Equations (15) and (16). I think you could choose and .
- Page 8, line 189: “The performance of PNN is much better than other classifier”, please support your statement. What are the reasons that led you to this idea?
- Page 9, lines 208-209: why did you choose the “F”, “W” and “H” as notation, they are hard to follow. I suggest you explain why did you choose them or to select another set of notations, more intuitive.
- Page 11, figure 8: I suggest you add to the title “with FWH”, as “Accuracy vs. SNR for each signals using FWH approach”
- Please recall the exact steps (and chosen parameters) you used in your proposed algorithm in order for the results to be reproductible
Minor concerns:
- Page 1, line 6: “…features in different domain…” is “…features from different domain…”
- Page 1, line 20: the keyword “PNN neural network” should be just “probabilistic neural network”
- Page 2, line 46: “…classification methods is mainly based…” is “…classification methods are mainly based”
- Page 4, line 111: please remove “been” from “information been preserved”.
- Page 4, line 113: “stacked auto encoder” replace with SAE
- Page 6, Figure 4: the third column is for 4FSK and not for “44SK”
- Page 9, line 205: “Such large a dynamic input…” is “Such a large dynamic input…”
- Page 9, table 3, first parameter: “sapling frequency” in stead of “sampling frequency”
- Page 9, line 207: rewrite ”…we compare the the performances…” as ”…we compare the performances…”
- Page 11, line 238: rewrite “…when …” as “…when …”
- Page 11, rewrite the Ox label of Figure 8: “SNR/dB” as “SNR (dB)”. Also, for Figure 6 and Figure 9.

Reviewer 3 Report
The paper addressed the problem of AMC with machine learning methods. The topic itself is interesting and worth being studied. However, there are some issues hindering the readers from understanding the contributions of the authors. The authors are suggested to consider the following aspects:
(1) The mathematical expressions given in the paper are not rigorous. Especially, it is difficult to know the data size in Equations 5, 6 , and 7. Understanding the relationship between the received signal and the data samples processed in the SAE is not easy. A few explanations of symbols or parameters, e.g. \zeta, d_\zeta, and C_{pq} in page 8, are not following a professional way.
(2) The notations of the parameters are not consistent. For instance, w and W are used to denote the same weighting matrix. What is \rho and \rho_i? n and Ns both stand for the number of samples? What are 'dx' and 'dy' in page 5? In line 180, why not just use N_s^2 to denote the square of N_s? The authors may need to be more careful in handling the mathematical issues.
(3) There are some non-negligible mistakes in the paper. In line 26, how could a more complex modulation result in a high level noise? Does noise come from the modulation amplitude or order? In line 27 and 202, what does the 'dBHz' stand for?
(4) As for the contributions of the paper, the authors claimed that the proposed scheme is more robust in a high dynamic range. However, since there are already various AMC algorithms in the literature, we see no comparison of the proposed scheme and the existing methods. In addition, the signal model described in equation (1) seems too simple for AMC.
(5) There are too many grammar mistakes throughout the paper. The English writing of this paper is lacking a quality that can be accepted.
Round 2
Reviewer 2 Report
This paper presents an AMC framework based on deep learning and highlights the possibility to fuse different features extracted from the input signals in order to achieve superior performances over the methods that are already in used nowadays. The proposed method is validated by simulation results. The paper is well written and deserves to be published.